# Calpain-Independent Intracellular Protease Activity Is Elevated in Excitotoxic Cortical Neurons Prior to Delayed Calcium Deregulation and Mitochondrial Dysfunction

**DOI:** 10.3390/biom12071004

**Published:** 2022-07-20

**Authors:** Brian M. Polster, Karla A. Mark, Rafael Arze, Derek Hudson

**Affiliations:** 1Buck Institute for Age Research, Novato, CA 94945, USA; karlamrk@yahoo.com; 2Department of Anesthesiology and Center for Shock, Trauma and Anesthesiology Research (STAR), University of Maryland School of Medicine, Baltimore, MD 21201, USA; 3Biosearch Technologies, Inc., Novato, CA 94949, USA; rafaelarze49@gmail.com (R.A.); dhudsonhome@comcast.net (D.H.)

**Keywords:** calpain, glutamate, NMDA, excitotoxicity, proteasome, mitochondria, membrane potential, neurodegeneration, MG132, E64d

## Abstract

Glutamate excitotoxicity contributes to many neurodegenerative diseases. Excessive glutamate receptor-mediated calcium entry causes delayed calcium deregulation (DCD) that coincides with abrupt mitochondrial depolarization. We developed cA-TAT, a live-cell protease activity reporter based on a vimentin calpain cleavage site, to test whether glutamate increases protease activity in neuronal cell bodies prior to DCD. Treatment of rat cortical neurons with excitotoxic (100 µM) glutamate increased the low baseline rate of intracellular cA-TAT proteolysis by approximately three-fold prior to DCD and by approximately seven-fold upon calcium deregulation. The glutamate-induced rate enhancement prior to DCD was suppressed by glutamate receptor antagonists, but not by calpain or proteasome inhibitors, whereas DCD-stimulated proteolysis was partly attenuated by the proteasome inhibitor MG132. Further suggesting that cA-TAT cleavage is calpain-independent, cA-TAT fluorescence was observed in immortalized *Capn4* knockout fibroblasts lacking the regulatory calpain subunit. About half of the neurons lost calcium homeostasis within two hours of a transient, 20 min glutamate receptor stimulation. These neurons had a significantly (49%) higher mean baseline cA-TAT proteolysis rate than those maintaining calcium homeostasis, suggesting that the unknown protease(s) cleaving cA-TAT may influence DCD susceptibility. Overall, the results indicate that excitotoxic glutamate triggers the activation of calpain-independent neuronal protease activity prior to the simultaneous loss of calcium homeostasis and mitochondrial bioenergetic function.

## 1. Introduction

Glutamate is the major excitatory neurotransmitter of the brain. Glutamatergic signaling through calcium (Ca^2+^)-permeable N-methyl-D-aspartate (NMDA) receptors is thought to be crucial for learning and memory [1]. However, excess glutamate release occurs following traumatic brain injury or stroke, and in various other neurodegenerative conditions, triggering a type of cell death called excitotoxicity [2]. Excitotoxic cell death is preceded by loss of intracellular Ca^2+^ homeostasis and is dependent on NMDA receptor-mediated Ca^2+^ influx [3,4,5].

Mitochondria, negatively charged organelles which produce ATP to meet cellular energy needs [6], take up Ca^2+^ through the mitochondrial calcium uniporter (MCU) channel [7,8]. This Ca^2+^ buffering by mitochondria helps maintain tight control over intracellular Ca^2+^ levels, enabling Ca^2+^ to continue to regulate diverse processes throughout the cell following the cessation of glutamate receptor stimulation [5]. When the level of glutamate receptor stimulation is abnormally high and/or prolonged, mitochondrial Ca^2+^ uptake capacity and other intracellular Ca^2+^ buffering mechanisms are exceeded. This results in loss of intracellular Ca^2+^ homeostasis, accompanied by nearly complete and persistent mitochondrial depolarization [5].

Ca^2+^ deregulation is observed in primary rodent neuronal cell cultures treated with an excitotoxic concentration of glutamate even after resting Ca^2+^ is restored by the addition of glutamate receptor antagonists [5]. The timing of this so-called delayed calcium deregulation (DCD) in individual neurons is stochastic. The delay in Ca^2+^ deregulation indicates that the initial NMDA receptor-mediated Ca^2+^ rise triggers intracellular events leading to inevitable Ca^2+^ homeostasis collapse despite the cessation of Ca^2+^ influx. A better understanding of the intracellular signaling events that commit neurons to DCD and resultant death following transient excitotoxic glutamate receptor stimulation is important because it may lead to novel, neuron-sparing intervention strategies.

Previously, we hypothesized that activation of Ca^2+^-dependent calpain proteases promote mitochondrial dysfunction and DCD following glutamate-mediated Ca^2+^ entry [9]. However, we [9] and others [10] failed to detect cytoplasmic calpain activity until after DCD when using α-spectrin substrate-based reporters. Nevertheless, the hypothesis that Ca^2+^-activated protease activity contributes to the delayed mitochondrial dysfunction and DCD remained attractive because apoptosis-inducing factor (AIF), a mitochondrial protein reported to causatively contribute to excitotoxic cell death upon nuclear translocation [11,12], is released from the mitochondrial inner membrane by calpain- [13,14] or cathepsin-mediated [15] proteolysis.

Here, we employed cA-TAT, a second-generation version of a cell-permeable, fluorogenic reporter based on a calpain cleavage sequence within a different calpain substrate, vimentin [16], to revisit the question of whether cytoplasmic protease activity in cortical neurons increases prior to DCD. We found that excitotoxic glutamate exposure causes an immediate increase in the rate of intracellular protease reporter cleavage that appears to be Ca^2+^-dependent. However, unexpectedly, this increase was insensitive to inhibitors of both calpains and the proteasome. A significantly higher baseline cA-TAT proteolysis rate was observed in the fraction of neurons undergoing DCD during the time course of our experiment, raising the important possibility that the unknown protease or proteases primarily responsible for cA-TAT processing contribute to the delayed perturbations in Ca^2+^ and mitochondrial homeostasis.

## 2. Materials and Methods

### 2.1. Reagents

Cal Fluor Orange-GSGTSS-K(BHQ-2) amide (cA) and Biotin-BAla-(K-CFO560)-GSGTSS-K(BHQ-2)-BAla-GRKKRRQRRRPQ-amide (cA-TAT) were synthesized by Biosearch Technologies, Inc. (Novato, CA, USA) [1]. Tetramethylrhodamine methyl ester (TMRM^+^), Fluo-4FF, Neurobasal medium, B27 supplement, and GlutaMAX were obtained from Invitrogen (Carlsbad, CA, USA). Ionomycin was obtained from EMD Biosciences (San Diego, CA, USA). Streptavidin-horseradish peroxidase (HRP) was obtained from Life Technologies-Thermo Fisher (Carlsbad, CA, USA). Other reagents were obtained from Sigma-Aldrich (St. Louis, MO, USA).

### 2.2. Preparation and Culture of Primary Neurons

Primary cortical neurons were prepared from 1–2 pairs of E18 rat cortices obtained from BrainBits™, LLC (Springfield, IL, USA) as previously described [9,16]. Briefly, brain tissue was enzymatically dissociated using papain, triturated gently, and then plated at a density of 1 × 10^5^ cells/well in poly-D-lysine-coated Lab-Tek 8-well chamber slides (Nunc). Fetal bovine serum (FBS, 1%) was included in the Neurobasal culture medium described below, but only during the initial plating to facilitate attachment to the chamber coverglass. Cells were subsequently maintained at 37 °C in Neurobasal medium containing B27 supplement (2%), GlutaMax (0.5 mM), penicillin (100 IU/mL), and streptomycin (100 µg/mL) for 14–15 days in vitro (DIV) prior to experiments, with half medium changes performed every 3–4 days. A low-oxygen incubator with a humidified atmosphere of 92% N_2_/5% CO_2_/3% O_2_ was used for culture to prevent oxidative stress due to supraphysiological O_2_ [9,16], as brain tissue oxygen level (5–45 mm Hg, 1–6% O_2_) is substantially lower that the oxygen level in the atmosphere (160 mm, 21% O_2_) [17]. Immediately prior to live-cell imaging experiments, the Neurobasal culture medium was removed, and the neurons were gently washed twice with and incubated in imaging medium. The imaging medium consisted of 120 mM NaCl, 3.5 mM KCl, 1.3 mM CaCl_2_, 0.4 mM KH_2_PO_4_, 20 mM N-Tris-(hydroxymethyl)-methyl-2-amino-ethanesulfonic acid, 5 mM NaHCO_3_, 1.2 mM Na_2_SO_4_, and 15 mM D-glucose, pH 7.4.

### 2.3. Culture of Mouse Embryonic Fibroblasts (MEFs)

Immortalized *Capn4* knockout (*Capn4*^−/−^) MEFs and wild-type (*Capn4*^+/+^) MEF control cells were generously provided by Dr. Peter Greer (Queen’s University, Kingston, ON, Canada) [18]. The cells were cultured in Dulbecco’s modified Eagle medium (DMEM) supplemented with 10% FBS, L-glutamine (2 mM), penicillin (100 IU/mL), and streptomycin (100 µg/mL). Fibroblasts were maintained at 37 °C in a humidified atmosphere of 95% air/5% CO_2_.

### 2.4. Live-Cell Imaging

An LSM™ 5 Pascal laser scanning confocal system (Carl Zeiss AG, Oberkochen, Germany) was used to simultaneously image changes in intracellular protease activity and calcium level in a temperature-controlled room air enclosure at 37 °C with the fluorescent probes cA (5 µM) or cA-TAT (1 µM) and Fluo-4FF (0.5 μM), respectively. The probe cA is an internally quenched protease substrate containing a calpain cleavage site of vimentin [16] (Figure 1A) and cA-TAT is a modified version with superior cell permeability (Figure 1B). Fluo-4FF is a cell-permeable, low-affinity calcium indicator (K_d_~9.7 μM) that fluoresces intensely only in neurons displaying loss of calcium homeostasis [9,19]. The imaging set-up consisted of an Axiovert 100 M inverted microscope equipped with a 20x NEOFLUAR NA 0.5 air objective and argon (488 nm) and helium/neon (543 nm) lasers.

Neurons in eight-well Lab-Tek chambers were pre-treated with calpain inhibitor (MDL28170 or E64d, each at 20 μM), proteasome inhibitor MG132 (20 µM), or dimethyl sulfoxide (DMSO) vehicle control for 10–30 min, as indicated in figure legends, and then cA or cA-TAT was loaded for 80 min prior to image acquisition. In cA-TAT imaging experiments, the extracellular protease probe was removed by replacing the imaging medium with fresh imaging medium containing Fluo-4FF and DMSO or inhibitor, but no cA-TAT. The cA protease reporters were excited at 543 nm and emitted fluorescence was collected between 560 and 615 nm, while the excitation and emission wavelengths for Fluo-4FF imaging were 488 nm and 505–530 nm, respectively.

For experiments assessing the timing of mitochondrial membrane potential loss with respect to Ca^2+^ deregulation, neurons were loaded with Fluo-4FF, as above, but in conjunction with TMRM^+^ (100 nM) instead of cA-TAT. The dye-loading period for TMRM^+^, used together with Fluo-FF, was 2 h at 37 °C. TMRM^+^ was excited at 543 nm and emissions were monitored using a 560LP filter, while Fluo-4FF fluorescence was simultaneously monitored using the excitation/emission wavelengths described above. Images were acquired at four-minute intervals for the duration of each experiment, using the Multi-Time Lapse Module to acquire fluorescence time-course measurements from four independent wells in parallel.

No crosstalk between either the emissions of cA or cA-TAT and Fluo-4FF or the emissions of TMRM^+^ and Fluo-4FF was detected. Regions of interest corresponding to all neuronal somas within imaged fields acquired from 2–4 independent wells of 1–2 primary neuronal preparations were selected for the quantification of fluorescence over time. All protease inhibitors were tested on ≥3 independent wells of two or more different primary neuronal preparations with similar results. However, because there were some differences in protocol (e.g., the concentration of cA-TAT probe, or the use of cA vs. cA-TAT), only exact replicates were combined for analysis. Rates of cA-TAT proteolysis were determined by linear regression analysis. A cell was excluded from analysis if: (1) the noise-to-signal ratio precluded either a linear fit of the baseline rate used for normalization or an ionomycin response (suggestive of poor probe loading); (2) the cell underwent lysis and lost the fluorescent indicator before any post-baseline rate could be calculated; (3) the cell spontaneously underwent Ca^2+^ deregulation prior to glutamate addition; and (4) the cell recovered calcium homeostasis following DCD (a rare event). These exclusion criteria are discussed further in Appendix A, with examples shown in Figure A1.

### 2.5. Gel Electrophoresis and Endogenous Protein or Biotin Detection

Neuronal lysates from cA-TAT-loaded cells were harvested in 50 µL of radioimmunoprecipitation assay (RIPA) buffer containing protease inhibitor cocktail following the acquisition of fluorescent time-courses and then subjected to sodium dodecyl sulfate-polyacrylamide gel electrophoresis (SDS–PAGE), as previously described [9]. Proteins were transferred to polyvinylidene difluoride (PVDF) membranes (Bio-Rad, Hercules, CA, USA) and, after one-hour blocking in 5% non-fat milk, the blots were incubated at 4 °C overnight in primary antibody to detect α-spectrin and β-actin. The primary antibody solutions used were a 1:500 dilution of mouse monoclonal anti-spectrin (nonerythroid) alpha chain (Clone AA6, Sigma-Aldrich, RRID:AB_94295) and a 1:5000 dilution of mouse anti-β-actin monoclonal antibody (Clone AC-74, Sigma-Aldrich, RRID:AB_476743), respectively. After three 10 min washes in Tris-buffered saline containing 0.05% Tween-20 (TBS–T), blots were incubated in 1:20,000 of anti-mouse HRP-conjugated secondary antibody for 45 min at room temperature, followed by three additional 10 min TBS–T washes. Bands were visualized using SuperSignal West Pico or West Fempto enhanced chemiluminescence substrates (Thermo Fisher Scientific), followed by exposure using X-ray film. Biotin detection was performed by incubating the PVDF membrane with 1:20,000 streptavidin–HRP overnight at 4 °C immediately after the one-hour blocking step, followed by three TBS–T washes and exposure using X-ray film, as described above.

### 2.6. Statistical Analysis

Statistical analysis was performed using GraphPad Prism version 9.3 (San Diego, CA, USA). A Welch’s unequal variances *t*-test was used to test for an effect of protease inhibitor treatment on baseline cA-TAT proteolysis rate and, also, to compare the baseline rate in cells that did or did not undergo Ca^2+^ deregulation in response to glutamate. We were unable to use repeated measures analysis of variance (ANOVA) to evaluate whether the acute treatments (i.e., glutamate, MK-801+NBQX, and ionomycin) changed the rate of cA-TAT proteolysis in individual cells because a few cells underwent lysis before the ionomycin rate could be calculated, and the repeated measures ANOVA cannot handle missing values. Therefore, the data were analyzed in GraphPad Prism by implementing the mixed-effects model using the Restricted Maximum Likelihood (REML) method, which employs a compound symmetry covariance matrix. Geisser–Greenhouse correction was used. This method gives the same *p*-values and multiple comparisons tests as repeated measures ANOVA when there are no missing values. Given the assumption that the missing ionomycin rate values resulting from the rare lysis events (3–4 cells per condition) were random, i.e., unrelated to the level of protease activity, the results can be interpreted in the same way. Finally, to test for inhibitor effects on the baseline-normalized rates measured following glutamate, MK-801+NBQX, or ionomycin addition, we used a two-way ANOVA followed by Šídák’s multiple comparisons test. *p*-values < 0.05 were considered significant.

## 3. Results and Discussion

### 3.1. Glutamate Stimulates the Rate of cA Fluoresescent Product Accumulation Prior to DCD in Calpain Inhibitor-Insensitive Fashion

The internally quenched fluorogenic protease substrate cA consists of the Cal Fluor Orange^TM^ fluorophore and BHQ-2^TM^ quencher separated by a calpain cleavage site derived from the structural protein vimentin (Figure 1A) [16]. Previously, we showed that cA is cleaved by purified erythrocyte calpain-1 [16]. We found that the intact molecule, but not the fluorescent cleavage product, is cell-permeable, and that intracellular fluorescence accumulates in hippocampal neurons incubated in nominally Ca^2+^-free imaging medium when the Ca^2+^ ionophore ionomycin combined with exogenous Ca^2+^ is added [16]. In cell-free assays, the rate of calpain-mediated cA digestion was more than five-fold slower than an “optimized” calpain substrate, cXe, but it was very resistant and moderately resistant to proteolysis by trypsin and chymotrypsin, respectively [16]. cA’s predicted resistance to non-specific degradation by the trypsin- and chymotrypsin-like activities of the proteasome and its sensitivity to Ca^2+^ entry in neurons led us to evaluate whether the cA reporter could be used to image changes in the activity of calpains or other intracellular proteases following excitotoxic glutamate exposure.

Addition of cA to primary rat cortical neurons matured for 14–15 days in vitro led to a low, variable extent of baseline fluorescent product accumulation in the cell bodies of healthy neurons maintaining calcium homeostasis (Figure 2A, panels i and ii). Exposure to excitotoxic glutamate (100 µM) for 40 min caused delayed calcium deregulation in many cells, as indicated by the bright green fluorescence of the low-affinity Ca^2+^ indicator Fluo-4FF (Figure 2A, panel iii). The glutamate treatment also resulted in a great enhancement in the cA signal in many cells (Figure 2A, panel iv). Upon overlay of the Fluo-4FF signal with cA fluorescence, Ca^2+^-deregulated cells generally appeared yellow, suggesting that extensive cleavage of the protease reporter molecule to the non-permeant fluorescent product had occurred (Figure 2A, panel iv, arrows). However, red-appearing cells with low Fluo-4FF signal were also observed in the overlay, indicating that protease reporter signal enhancement occurred without Ca^2+^ homeostasis loss in some neurons (Figure 2A, panel iv, arrowheads). Unexpectedly, excitotoxic glutamate treatment resulted in the appearance of neurons with enhanced cA fluorescence both before and after DCD even in the presence of the calpain inhibitor MDL28170 (Figure 2B). MDL28170 reduced the appearance of the calpain-generated 150 kilodalton α-spectrin cleavage fragment while, as expected [20], increasing the level of the caspase-generated 120 kilodalton fragment (Figure 2C), confirming effective calpain inhibition. These results raised the possibility that one or more proteases other than calpain are responsible for cA proteolysis in cells.

### 3.2. Sustained Mitochondrial Depolarization Occurs Simultaneously with Ca^2+^ Deregulation

We were interested in determining whether glutamate-stimulated protease activity occurs upstream of mitochondrial bioenergetic dysfunction, but the mitochondrial membrane potential-sensitive probe tetramethyl rhodamine methyl ester (TMRM^+^) could not be imaged in conjunction with cA owing to spectral overlap. Therefore, we sought to confirm that irreversible mitochondrial depolarization occurs simultaneously with Ca^2+^ deregulation in our cortical neuron cultures continuously exposed to excitotoxic glutamate (as observed by others in similar excitotoxicity models [21,22,23]), so that the time of DCD could be used as a proxy for the time of mitochondrial depolarization. To accomplish this, we performed imaging experiments with the fluorescent indicators TMRM^+^ and Fluo-4FF, using TMRM^+^ at a concentration (100 nM) that results in self-quenching in the mitochondrial matrix [21]. When TMRM^+^ is used in “quench mode,” mitochondrial depolarization results in efflux of TMRM^+^ from the matrix and consequent dequenching of its fluorescence emission [21]. The result is a transient spike in cytoplasmic TMRM^+^ fluorescence prior to the dye’s subsequent escape across the plasma membrane, revealing the timing of mitochondrial depolarization with respect to DCD when used in conjunction with Fluo-4FF.

Figure 3A shows quantification of TMRM^+^ fluorescence over time in regions of interest corresponding to neuronal somas from 10 representative neurons within an imaged well. We saw a relatively small TMRM^+^ fluorescence increase in the majority of cells immediately following glutamate addition (Figure 3A and representative Cell B in Figure 3B), consistent with that reported to result from ionotropic glutamate channel-mediated plasma membrane depolarization [21]. However, in a few cells showing immediate Ca^2+^ deregulation (ICD), a much larger increase in TMRM^+^ fluorescence was observed, followed by a rapid drop below baseline (Figure 3A and representative Cell A in Figure 3B). These large transient TMRM^+^ fluorescence “spikes” are consistent with mitochondrial efflux-associated dequenching upon mitochondrial membrane potential depolarization. In neurons initially showing the smaller increase attributable to plasma membrane depolarization, a similar larger fluorescence spike occurred later with variable timing, which was followed by a rapid and sustained drop in TMRM^+^ signal to below baseline (Figure 3A,B, representative Cell B). The timing of TMRM^+^ fluorescence spike invariably coincided with intracellular Ca^2+^ deregulation (Figure 3B; data not shown), regardless of the time when loss of Ca^2+^ homeostasis occurred.

### 3.3. Description of the Second-Generation Fluorogenic Protease Reporter cA-TAT

Our findings up to this point, in combination with previous findings of cytoplasmic calpain activation only after DCD [9,10], suggest that glutamate stimulates the activity of protease(s) other than calpain prior to the onset of DCD and persistent mitochondrial depolarization. However, there are alternative explanations for our findings. One limitation of the experiments described by Figure 2 is the large cell-to-cell variation in baseline signal observed, which could be due to a variable extent of reporter accumulation and/or true cell-to-cell variability. A second limitation is that, because cA was continuously present in the imaging medium following its addition, we could not rule out the possibility that the observed fluorescence increases were due to a glutamate- and/or DCD-induced enhancement in cA cell penetration rather than an increased rate of intracellular proteolysis. Finally, although it is unlikely, based on the fluorophore-quencher design, it is possible that the fluorescence changes we saw in cells were caused by a dequenching conformational change in the intact cA molecule rather than by genuine proteolysis that liberates the fluorophore from the quencher.

Therefore, we designed a second-generation reporter to help interpret our initial findings. First, we added an arginine-rich HIV TAT transporter sequence GRKKRRQRRRPQ [24] to cA as a C-terminal extension, with the goal of achieving enhanced and more uniform cell penetration (Figure 1B). Second, we added an N-terminal biotin tag to cA, which enabled a secondary method of cleavage product detection by exploiting the strong affinity between biotin and a labeled streptavidin molecule (Figure 1B). To eliminate the possibility of fluorescence increases due to an increased rate of reporter entry, in the subsequently described experiments the imaging medium was replaced with cA-TAT-free medium following an 80 min reporter-loading period to remove non-internalized protease substrate.

We found that although the TAT sequence addition allowed us to reduce the probe concentration used in imaging experiments by five-fold, from 5 to 1 µM, high cell-to-cell variability in baseline cA fluorescence levels was still observed. Images of cortical neurons loaded with cA-TAT are shown before and after glutamate addition in Figure 4A. Similar to the results with cA, red fluorescence was highest in cells that underwent DCD. However, some neurons showed fluorescence increases following glutamate addition without loss of Ca^2+^ homeostasis during the time course of the experiment.

To investigate whether the fluorescence changes observed in glutamate-treated cells correlated with cA reporter proteolysis as detected by Western blotting, neurons were lysed following imaging and subjected to SDS–PAGE. A faster-migrating band relative to the parent molecule was detected in lysate from glutamate-treated cells (Figure 4B). Cell-to-cell variability in cA-TAT accumulation, well-to-well variability in neuronal cell number, and baseline cA-TAT proteolysis in the absence of glutamate precluded quantitative comparisons of protease activity by Western blotting in bulk cell populations. Nevertheless, the finding of a band of a size consistent with the predicted cleavage product provides additional evidence that the appearance of red fluorescence in the cell soma reflects bona fide intracellular cA proteolysis.

### 3.4. Evidence That Glutamate-Induced cA-TAT Fluorescence Accumulation Is Due to Calpain-Independent Cleavage

Next, we sought to obtain additional evidence for calpain-independent cA-TAT cleavage. Although 15 mammalian calpain protease family members have been reported [25], calpain-1 and calpain-2 are highly expressed by neurons and are the most studied in the context of neurodegeneration [26,27]. Each protease consists of a unique large catalytic subunit and a common small regulatory subunit encoded by the *Capn4* gene. *Capn4* is required for the activity of both these calpains, as well as for the activity of most of the other family members [25]. As complete *Capn4* knockdown by small interfering RNA in primary cultures of mature cortical neurons is difficult to achieve, we tested whether Ca^2+^ ionophore-induced cA-TAT fluorescent product accumulation could be observed in previously described immortalized *Capn4* knockout (^−/−^) mouse embryonic fibroblasts [18]. We found that both *Capn4*^−/−^ MEFs and wild-type MEF control cells exhibited detectable intracellular red fluorescence upon baseline cA-TAT incubation and notable intracellular red fluorescence following ionomycin treatment (Figure 4C). A limitation of this experiment was that the downstream proteases activated by Ca^2+^ in ionomycin-treated MEF cells may have differed from those in glutamate-treated neurons because NMDA receptor-mediated Ca^2+^ entry exhibits a privileged access to mitochondria [28]. A detailed quantitative analysis was not performed, as the focus of our study is on glutamate-treated neurons. Nevertheless, these data support the hypothesis that cA-TAT is processed intracellularly by a protease (or proteases) other than calpain-1 or -2.

Members of the cysteine class of cathepsin proteases have been detected in isolated mitochondria, and cathepsins B, L, and S can cleave AIF to the same-sized truncated C-terminal fragment that is generated by calpain-1 [14,15]. Therefore, we tested the ability of E64d, a cell-permeable, irreversible cysteine protease inhibitor, to suppress the rate of cA-TAT cleavage in glutamate-exposed cortical neurons, either upstream or downstream of DCD. To quantitatively compare proteolysis rates in the absence and presence of the inhibitor, regions of interest corresponding to neuronal cell bodies were generated and cA-TAT and Fluo-4FF fluorescence values acquired in parallel from multiple wells at four-minute intervals were plotted. Figure 4D shows the fluorescence time courses in a representative neuron undergoing DCD approximately 30 min after glutamate addition, with the blue line illustrating the increased slope of cA-TAT fluorescence accumulation immediately following glutamate exposure. The majority of imaged neurons underwent DCD over the two-hour time course of the experiment when either vehicle (DMSO) or E64d was present, although a slightly higher percentage was observed with E64d (Table 1).

The baseline cA-TAT proteolysis rate showed high variability and was not significantly altered by E64d (Figure 5A and Table 2). Therefore, the cA-TAT proteolysis rates following glutamate addition and spontaneous DCD were normalized to the baseline rate in individual cells (Figure 5B). Since the time resolution was not high and we were primarily interested in the effect of Ca^2+^ deregulation on protease activity (regardless of when it occurred), cells showing Ca^2+^ deregulation at the first time point (4 min) after glutamate addition were classified as DCD rather than considered separately as ICD. Furthermore, since so few cells maintained Ca^2+^ homeostasis in the presence of E64d (Table 1), we tested for an effect of E64d on the glutamate- and DCD-stimulated rates only in the DCD subpopulation. The results of the analysis indicated that neither the initial cA-TAT proteolysis rate following glutamate addition nor the proteolysis rate measured once spontaneous DCD occurred were significantly altered by E64d (Figure 5B). As calpains are among the cysteine proteases inhibited by E64d, these results further support the conclusions of the MDL28170 and *Capn4^−/−^* experiment that enzyme(s) other than calpain are responsible for the digestion of cA reporters in cells.

### 3.5. Glutamate-Stimulated cA-TAT Proteolysis Is Acutely Inhibited by Glutamate Receptor Antagonists

Although multiple lines of evidence indicate that cA or cA-TAT proteolysis is independent of Ca^2+^-dependent calpain proteases, the cA fluorescence accumulation in cells behaves as if it is Ca^2+^-dependent. This behavior includes: (1) an increased rate of fluorescence accumulation upon glutamate addition (which stimulates NMDA receptor-mediated Ca^2+^ entry), (2) increased fluorescence accumulation upon spontaneous DCD, and (3) direct stimulation of fluorescence accumulation rate by Ca^2+^ ionophore-mediated Ca^2+^ entry. Therefore, we asked whether the glutamate-stimulated cA-TAT accumulation rate could be acutely reversed by blocking pathways of Ca^2+^ entry. To accomplish this, we added MK-801 to block NMDA receptor-mediated Ca^2+^ influx and, at the same time, the AMPA receptor antagonist NBQX to suppress glutamate-mediated plasma membrane potential depolarization. The latter should largely antagonize Ca^2+^ entry through voltage-dependent Ca^2+^ channels, while also preventing entry through the Ca^2+^-permeable AMPA receptor subclass. We found that the MK-801 plus NBQX combination not only significantly inhibited the glutamate-stimulated proteolysis rate, but also significantly suppressed the cA-TAT proteolysis rate to below baseline (Figure 4E and Table 3). These data further support the possibility that the protease activity responsible for cA-TAT digestion is Ca^2+^-dependent. Additionally, these results suggest that the cell-to-cell baseline cleavage rate variability may be, at least in part, reflective of variability in the endogenous level of neuronal activity (i.e., basal glutamate signaling). This is because the spontaneous electrical activity exhibited by mature cortical neurons in culture is expected to alter the basal levels of intracellular Ca^2+^ [29].

### 3.6. DCD- and Ionomycin-Stimulated cA-TAT Proteolysis, but Not Glutamate-Stimulated Proteolysis, Are Partly Attenuated by Proteasome Inhibition

Yet another challenge in identifying the cA-TAT-degrading protease(s) is the possibility of redundancy. The cA reporter was marginally susceptible to chymotrypsin in cell-free assays [16] and may be processed by the proteasome when calpains are inhibited. Thus, we next evaluated the proteosome inhibitor MG132, which at relatively high concentrations (≥10 μM) inhibits both the chymotrypsin-like activity of the proteasome and calpain enzymes, including the atypical calpain-5 that does not appear to require *Capn4* for activity [30,31]

By using the protocol employing MK-801 plus NBXQ addition to limit the period of glutamate receptor stimulation (Figure 4E), we were able to decrease the fraction of neurons undergoing DCD over a two-hour period (Table 4) compared with the E64d experiment (Table 1). This allowed us to also evaluate glutamate-stimulated cA-TAT proteolysis in cells that did not show DCD, with the Ca^2+^ ionophore ionomycin used as a positive control to confirm cA-TAT probe loading. MG132 (20 µM) pre-incubation did not affect the baseline cA-TAT proteolysis rate (Table 2). More cells underwent DCD in the presence of MG132 (45.9%) compared with its absence (28.7%), consistent with the proteasome’s known homeostatic role in damaged/misfolded protein removal [30]. Despite this, MG132 did not significantly affect the cA-TAT proteolysis stimulation observed immediately following excitotoxic glutamate exposure (Figure 5C and Table 3). However, we found that MG132 significantly reduced DCD-stimulated proteolysis in cells that underwent Ca^2+^ deregulation and ionomycin-stimulated proteolysis in non-deregulating cells to similar extents of ~45% (Figure 5C and Table 3). Although these results suggest that intracellular calcium deregulation enhances proteasome activity, a previous report suggested that NMDA receptor-mediated Ca^2+^ entry causes 26S proteasome disassembly in hippocampal neurons, decreasing proteasome activity following excitotoxic glutamate exposure [32]. Speculatively, the MG132 sensitivity of cA-TAT proteolysis observed exclusively after Ca^2+^ homeostasis loss may reflect an altered specificity of the 20S proteasome’s catalytic core once it is liberated from the 19S regulatory particle.

### 3.7. The Neuronal Subpopulation That Undergoes DCD Exhibits a Significantly Higher Baseline cA-TAT Proteolysis Rate

As noted earlier, MG132 did not significantly alter the baseline cA-TAT proteolysis rate. Therefore, to increase the statistical power so as to determine whether there is a relationship between the baseline level of cA-TAT proteolysis and the incidence of DCD, we combined the vehicle- and MG132-treated neurons from the last experiment. The average cA-TAT proteolysis rate (mean ± SEM) was ~50% greater in cells that underwent DCD within two hours of the start of the 20 min transient glutamate receptor stimulation compared with the neurons that were able to maintain Ca^2+^ homeostasis (1.81 ± 0.18, *n* = 124 vs. 1.20 ± 0.12, *n* = 109, *p* = 0.0066, Figure 5D). The magnitude of this difference may even be an underestimate, as the number of non-deregulating cells excluded from analysis due to low signal-to-noise ratio greatly outnumbered the number of DCD cells that were excluded based on the same criterion (Appendix A).

One hypothesis to explain this result is that the protease mediating most of the cA-TAT cleavage also promotes DCD following glutamate exposure by degrading protein substrates required for Ca^2+^ homeostasis. However, there are other possibilities. MK-801+NBQX dropped the cA-TAT proteolysis rate below baseline, suggesting that the baseline cA-TAT proteolysis rate reflects the endogenous level of glutamate receptor signaling. Therefore, either energy demand or another process associated with neuronal activity may underlie the correlation between high cA fluorescence accumulation rate and DCD occurrence. Alternatively, it may be the individual neuron’s ability to load cA-TAT probe and the pathways involved, rather than the ability to digest the reporter at a higher rate, that predicts DCD propensity. While the Western blot detection of biotin-conjugated cA-TAT cleavage product strongly indicates that intracellular fluorescence accumulation is due to reporter cleavage, further work is needed to understand whether the large cell-to-cell variability reflects differences in cA-TAT loading, true heterogeneity in protease activity, or some combination of the two.

We made initial attempts to identify the class of proteolytic activities responsible for cA-TAT digestion in cells. E64d insensitivity suggests that cA-TAT cleavage is not cysteine protease-mediated, ruling out calpains and several cathepsins, with additional pharmacological (MDL28170) and genetic (*Capn4^−/−^*) approaches supporting calpain-independence. We did observe significant inhibition by the proteasome inhibitor MG132, but only of the DCD- or ionomycin-induced cA-TAT proteolysis enhancement, and even then the inhibition was incomplete. This result suggests that multiple proteolytic activities contributed to cA-TAT cleavage over the time course of our imaging experiments.

The identity of the protease(s) responsible for the baseline and glutamate-stimulated rates, hypothesized to contribute to delayed Ca^2+^ deregulation, remain unknown. The lack of specificity and, in many cases, cell permeability of small-molecule protease inhibitors make it difficult to uncover the responsible proteases through additional pharmacology. Future experiments employing biochemical fractionation or genetic knockdown approaches may ultimately be required.

Finally, it is important to note that although cA-TAT cleavage appears to be Ca^2+^-dependent, the role of Ca^2+^ need not be direct. Activation of the Ca^2+^-dependent phospholipase PLA2G4A was recently reported to cause the translocation of a soluble lysosomal protease to the cytoplasm in rat cortical neurons and to promote lysosomal permeabilization in vivo following traumatic brain injury [33]. Lysosomes house numerous proteases, some of which exhibit altered activity at the neutral pH of the cytoplasm compared with the acidic pH of the lysosome [34,35]. Thus, it is possible that the modest cA-TAT fluorescence increase upon glutamate addition reflects activation of one or more proteases due to limited lysosomal permeabilization by Ca^2+^-dependent phospholipase, e.g., of a few lysosomes, whereas the larger increase upon DCD reflects more global lysosomal disruption. Future experiments with Ca^2+^-dependent phospholipase A2 inhibitors and lysosome-stabilizing agents could be carried out to test this hypothesis.

## 4. Conclusions

Using a novel, internally quenched fluorogenic peptide reagent, cA-TAT, this study demonstrated that excitotoxic glutamate exposure increases cortical neuron intracellular protease activity prior to delayed Ca^2+^ deregulation. Higher cA-TAT proteolysis rates in neurons showing DCD suggests that an unknown protease responsible for cA-TAT reporter cleavage influences DCD susceptibility. The identity of this protease (or proteases), its endogenous substrates, and whether it indeed contributes to mitochondrial dysfunction and DCD remain to be determined.

## Figures and Tables

**Figure 1 biomolecules-12-01004-f001:**
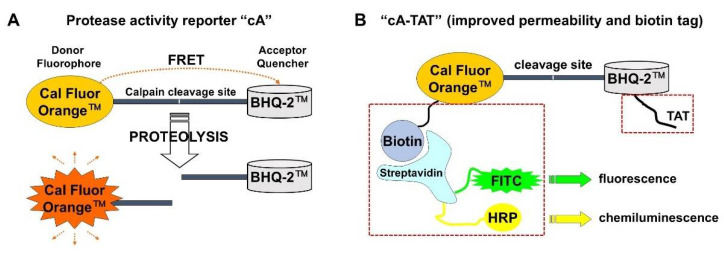
Schematics illustrating the design of the internally quenched small-molecule protease reporters used in this study. The diagram in (**A**) shows reporter cA, which consists of an N-terminal fluorophore, Cal Fluor Orange, and a C-terminal quencher, BHQ-2, separated by a peptide consisting of a calpain cleavage site within murine vimentin (GSG│TSS). The diagram in (**B**) shows the next-generation reporter cA-TAT, which includes the modifications highlighted by dashed boxes: an N-terminal biotin conjugation to cA and a C-terminal TAT peptide sequence (GRKKRRQRRRPQ). The biotin moiety enables detection of the intact molecule and the N-terminal cleavage product by streptavidin conjugated to either horseradish peroxidase (HRP) or a fluorescent label, such as fluorescein isothiocyanate (FITC), while the TAT peptide confers enhanced cell permeability.

**Figure 2 biomolecules-12-01004-f002:**
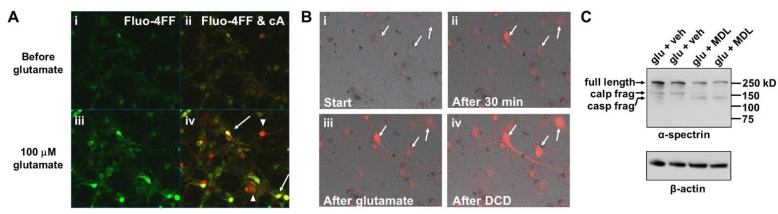
Excitotoxic glutamate (glu, 100 µM) treatment causes intracellular cA fluorescent product to accumulate prior to delayed Ca^2+^ deregulation (DCD), both in the absence (**A**) and presence (**B**) of the calpain inhibitor MDL28170 (MDL, 20 μM). In A, rat cortical neurons were simultaneously imaged for Fluo-4FF fluorescence (green, panel i) and cA fluorescence (red, merged with Fluo-4FF, panel ii) at 4 min prior to glutamate addition. Panels iii and iv in A show the same field of neurons imaged under the respective fluorescence channels at two hours post-glutamate exposure. In B, rat cortical neurons were imaged after a 30 min pre-incubation with MDL28170 and cA (panel i), imaged again after 30 min (panel ii, just prior to glutamate addition), imaged four minutes after glutamate exposure (panel iii), and imaged at two hours post-glutamate exposure (panel iv). The cA fluorescent signal (red) is shown as an overlay on the phase contrast images. The three cells indicated by arrows underwent DCD in the interval between when the images in panels iii and iv were taken, as identified by simultaneous Fluo-4FF imaging (not shown). (**C**) Western blot showing detection of the calpain substrate α-spectrin in cells pre-treated with MDL28170 or vehicle and then exposed to glutamate. Arrows denote full-length α-spectrin and calpain (calp)- and caspase (casp)-generated α-spectrin cleavage fragments (frag). β-actin was detected on the same membrane as a loading control.

**Figure 3 biomolecules-12-01004-f003:**
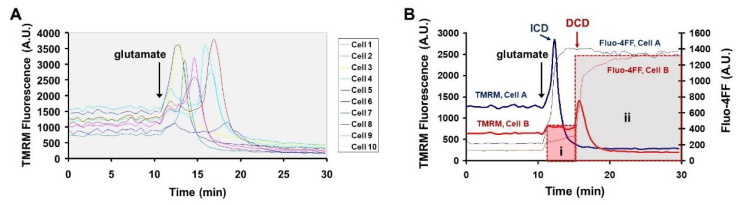
Mitochondrial depolarization timing in response to excitotoxic glutamate is stochastic and occurs simultaneous to Ca^2+^ deregulation. (**A**) TMRM^+^ fluorescence time courses from 10 representative individual neurons that were exposed to glutamate (100 µM) at the indicated time (arrow). (**B**) TMRM^+^ (left axis) and Fluo-4FF (right axis) values over time from two representative cells that underwent either immediate Ca^2+^ deregulation (ICD, Cell A, blue traces) or spontaneous delayed Ca^2+^ deregulation (DCD, Cell B, red traces) following glutamate addition. Continuous traces were plotted from images taken at 30 s intervals. Time intervals “i” (pink shading) and “ii” (gray shading) illustrate the periods before and after DCD, respectively, with the central question of this study being whether glutamate stimulates protease activity during the “i” period prior to DCD.

**Figure 4 biomolecules-12-01004-f004:**
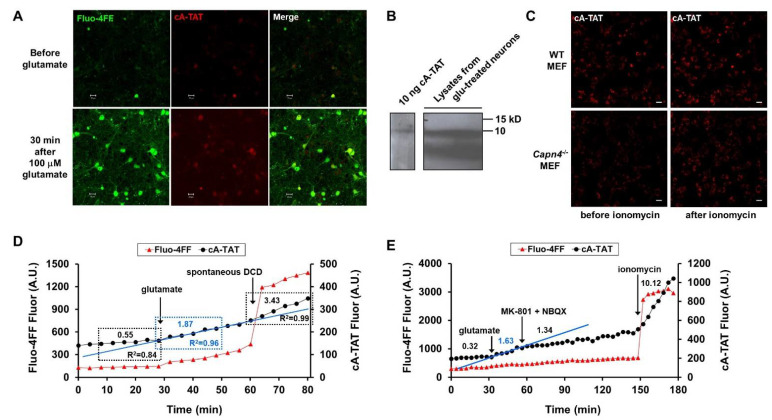
(**A**) Representative images of neurons loaded with Fluo-4FF (green) and cA-TAT (red) taken before and 30 min after glutamate (100 μM) treatment. (**B**) Streptavidin–HRP Western blot comparing the size of bands observed in two different lysates of glutamate-treated neurons to the band observed when the cA-TAT molecule was loaded directly onto the gel. (**C**) Images of cA-TAT fluorescence in wild-type (top) and *Capn4* knockout (^−/−^, bottom) immortalized mouse embryonic fibroblasts (MEFs) after 20 min of baseline recording following the 80 min loading period (left panels, “before ionomycin”) and following treatment with ionomycin (5 µM) for 40 min (right panels, “after ionomycin”). Scale bars in A and C are 20 µm. (**D**) Fluo-4FF (left axis) and cA-TAT (right axis) fluorescence (fluor) values over time for a representative neuron exposed continuously to glutamate (100 µM) that subsequently showed spontaneous delayed Ca^2+^ deregulation (DCD), as identified by the sudden jump in Fluo-4FF fluorescence (red trace). The dashed boxes highlight the data points for which the baseline, glutamate, and DCD rates were determined by linear regression analysis, with rates and R^2^ values given in each box. (**E**) Fluo-4FF (left axis) and cA-TAT (right axis) fluorescence values over time for a representative neuron sequentially exposed to glutamate (100 µM), a combination of the glutamate receptor antagonists MK-801 and NBQX to inhibit NMDA and AMPA-type receptors, respectively, and the Ca^2+^ ionophore ionomycin. The solid blue lines drawn in D and E highlight the increased slope of cA-TAT fluorescence immediately following glutamate addition. A.U., arbitrary units/min.

**Figure 5 biomolecules-12-01004-f005:**
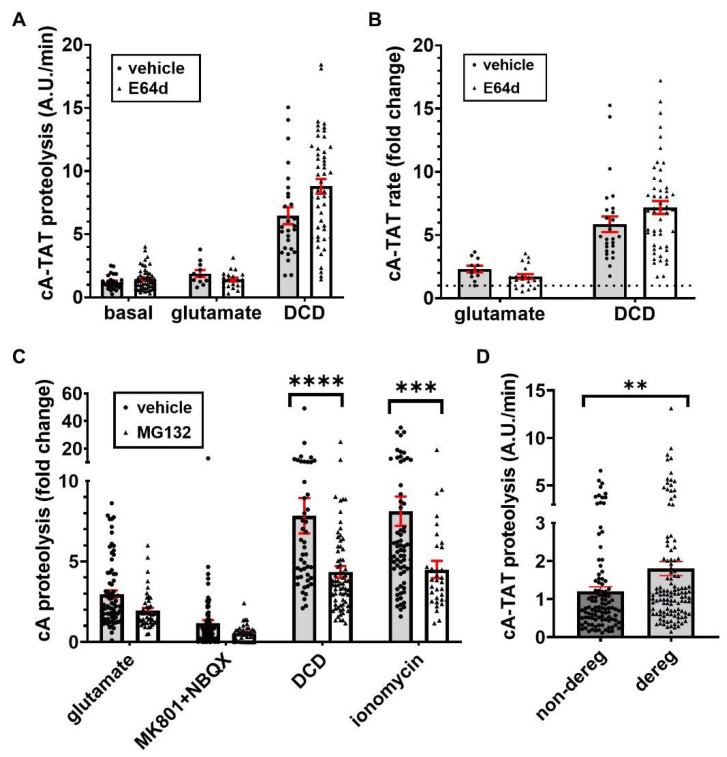
Indicated absolute (**A**) and baseline-normalized (**B**) cA-TAT rates in arbitrary units (A.U.)/min are shown in the presence of vehicle (0.1% DMSO) or cysteine protease inhibitor E64d (20 µM). In (**C**), the indicated baseline-normalized cA-TAT rates are shown in the presence of vehicle (0.2% DMSO) or protease inhibitor MG132 (20 µM). The graph in (**D**) compares the baseline cA-TAT proteolysis rates in cells that did or did not undergo Ca^2+^ deregulation within two hours of excitotoxic glutamate (100 µM) addition in the experiment described in C, regardless of the pre-treatment (vehicle or MG132). The data are shown as means ± standard errors of the means (SEMs), with the individual data points indicating individual cells. ** denotes *p* < 0.01, determined by Welch’s unequal variances test. *** and **** denote *p* < 0.001 and *p* < 0.0001, respectively, determined by two-way ANOVA followed by Šídák’s multiple comparisons test.

**Table 1 biomolecules-12-01004-t001:** Number (#) of cells undergoing delayed calcium deregulation (DCD) or maintaining Ca^2+^ homeostasis (non-dereg) in the absence and presence of the cysteine protease inhibitor E64d. Numbers in parenthesis denote the percentage of cells undergoing DCD for each treatment.

Treatment	Total Cell #	Non-Deregulating #	DCD #	DCD # Analyzed
Vehicle	69 ^1^	24	44 (63.8%)	27
E64d	63 ^1^	7	55 (87.3%)	51 ^2^

^1^ Includes one cell undergoing spontaneous Ca^2+^ deregulation prior to glutamate addition. ^2^ Includes one cell for which only the glutamate rate could be calculated due to cell lysis and consequent loss of probe, following DCD. See Appendix A for discussion of exclusion criteria and representative examples.

**Table 2 biomolecules-12-01004-t002:** Baseline cA-TAT proteolysis rates (arbitrary units/min) in the presence of the indicated treatments. Neither E64d nor MG132 significantly changed the proteolysis rate compared with their vehicle control (Welch’s unequal variances *t*-test; vehicles were 0.1% DMSO and 0.2% DMSO, respectively).

Treatment	Mean ± SD	Lower 95% CI	Upper 95% CI	*n*
E64d vehicle	1.19 ± 0.52	1.00	1.38	31
E64d	1.44 ± 0.85	1.20	1.67	52
MG132 vehicle	1.45 ± 1.95	1.09	1.81	115
MG132	1.60 ± 1.52	1.32	1.87	118

**Table 3 biomolecules-12-01004-t003:** cA-TAT proteolysis rates (arbitrary units/min) in neurons that did (Dereg) or did not (Non-dereg) undergo Ca^2+^ deregulation within two hours of glutamate addition when vehicle (0.2% DMSO) or the proteasome inhibitor MG132 (20 µM) was present. Values are given as means ± standard deviations, with the number of cells in parenthesis. Significance was evaluated by mixed-effects model ANOVA using REML.

Rate	Non-Dereg Vehicle	Non-Dereg MG132	Dereg Vehicle	Dereg MG132
Baseline	1.12 ± 1.34 (69)	1.34 ± 1.13 (40)	1.94 ± 2.56 (46)	1.73 ± 1.68 (78)
Glutamate	2.02 ± 1.52 (69) ^a,b^	2.05 ± 1.31 (40) ^a,b^	2.75 ± 2.09 (8)	2.81 ± 2.33 (5)
MK-801 + NBQX	0.69 ± 0.69 (69) ^a,b^	0.59 ± 0.79 (40) ^a,b^	1.57 ± 1.20 (7)	1.40 (2)
DCD	N/A	N/A	9.73 ± 9.99 (46)	6.09 ± 5.46 (78)
Ionomycin	6.33 ± 6.40 (66) ^a^	4.91 ± 3.63 (36) ^a^	N/A	N/A

^a^ *p* < 0.0001 compared with the respective baseline rate. ^b^ *p* < 0.0001 compared with the respective DCD or ionomycin rate.

**Table 4 biomolecules-12-01004-t004:** Cells undergoing delayed calcium deregulation (DCD) or maintaining Ca^2+^ homeostasis (Non-dereg) in the absence and presence of the proteasome inhibitor MG132 (*#* or %, as indicated). Numbers in parenthesis in the table refer only to the analyzed cells within each category.

Treatment	Total Cell #	DCD # (Analyzed)	Non-Dereg # (Analyzed)	% DCD (Analyzed)
Vehicle	174	50 (46)	124 (69)	28.7 (40.0)
MG132	205 ^1^	95 (78)	103 (40)	45.9 (66.1)

^1^ Includes four cells undergoing spontaneous Ca^2+^ deregulation prior to glutamate addition and three cells undergoing reversible Ca^2+^ deregulation. See Appendix A for discussion of exclusion criteria and representative examples.

## Data Availability

Data are contained within the article.

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
