# Peer review of "Calpain-Independent Intracellular Protease Activity Is Elevated in Excitotoxic Cortical Neurons Prior to Delayed Calcium Deregulation and Mitochondrial Dysfunction"

_biomolecules, 2022, doi:10.3390/biom12071004_

Round 1

Reviewer 1 Report

Manuscript Number:  biomolecules-1766730

Title: Calpain-independent Intracellular Protease Activity is Elevated 2 in Excitotoxic Cortical Neurons Prior to Delayed Calcium Deregulation and Mitochondrial Dysfunction

This study by Polster et al. shows that a calpain-independent unknown protease activity might contribute to DCD, which is of high interest in the field of excitotoxicity, since the contribution of calpains to excitotoxic neuronal death remains controversial. The manuscript is well written and clear, and most of the methods used are precise and appropriate. However, there are some issues that need to be addressed.

Major points:
-          The number of cultures assayed (n) is not specified and in some experiments (table 3 for example) the number of cells analyzed is extremely low.
-          All the evidence that cA-TAT proteolysis is calpain-independent comes from pharmacological approach. Thus, authors should show that the calpain inhibitors used in this model actually inhibit calpains, or give evidence of similar previous experiments using the same inhibitors.

Minor points:
-          Why authors show in Figure 2B phase contrast images and not Fluo-4FF images like in Figure 2A?
-          In Figure 3A the glutamate addition arrow is lacking.
-          A ionomycin treatment on MEFs may not be comparable to a glutamate addition to neurons, since the latter shows a privileged access to mitochondria and therefore downstream recruited proteases may not be the same at all. Authors should further discuss this point.
-          Figure 4C should show images of cA-TAT in control MEFs (before ionomycin) in order to see wether proteolysis is obvious.
-          Representative images of Fluo-4FF (Figure 4A) show a huge fluorescence increase after glutamate addition. However, the fluorescence increase showed by the traces in Figure 4D is very small and almost absent in Figure 4E. Authors should explain this discrepancy.

Reviewer 2 Report

in this study by using novel internally fluorogenic reagent, authors showed glutamate exposure increases cortical neurons, I thibk this is a priliminary 

results, authors should do further study to know themachanism of how this is happening and affecting mitochondria

Reviewer 3 Report

This article was designed to assess the correlation between calpain-activity, Ca2+ deregulation and mitochondrial disfunction, following the induction of cortical excitotoxicity. Glutamate-induced cytotoxicity of cortical neurons evokes an increase in intracellular Ca2+ that precedes cell death. In this work, the authors tested the hypothesis that Ca2+-dependent calpain proteases promote mitochondrial dysfunction and cell death. The study was carried out in a technically sound manner and it is clearly written. However, certain aspects of the study require further revisions. Below is a list of issues that I considered should be addressed before the manuscript is published in Biomolecules.

1)     Can the authors explain whether the ability of neurons to generate an immediate Ca2+ deregulation (ICD) or a delayed Ca2+ deregulation (DCD) correlates with differences in the basal level of TMRM fluorescence as presented in Fig. 2B. For example, we could speculate that the majority of cells with lower TMRM fluorescence presented DCD, whereas cells with high levels of TMRM fluorescence presented ICD due to differences in fluorophore distribution or other extrinsic factors.

2)     I wonder if the cultures of cortical neurons, used in this study are capable of generating spontaneous electrical activity, which can potentially alter the basal levels of intracellular Ca2+.

3)     The authors describe the effect of MK-801 and NBQX in limiting Ca2+ influx via the activation of NMDA and AMPA receptors. These two blockers significantly reduced glutamate induced proteolysis. Not sure if they can block Ca2+ influx via the activation of voltage-gated Ca2+ channels by using MK-801 and NBQX, rather that cobalt or cadmium ions.

Based on the results presented, it appears that Ca2+ dependent calpain activation is not involved in the Ca2+ and mitochondrial disfunction caused by glutamate. Can the authors speculate on the nature of the protease(s) involved in DCD?

Round 2

Reviewer 1 Report

All my points have been adequately addressed by the authors.